# G-Refine: A General Quality Refiner for Text-to-Image Generation

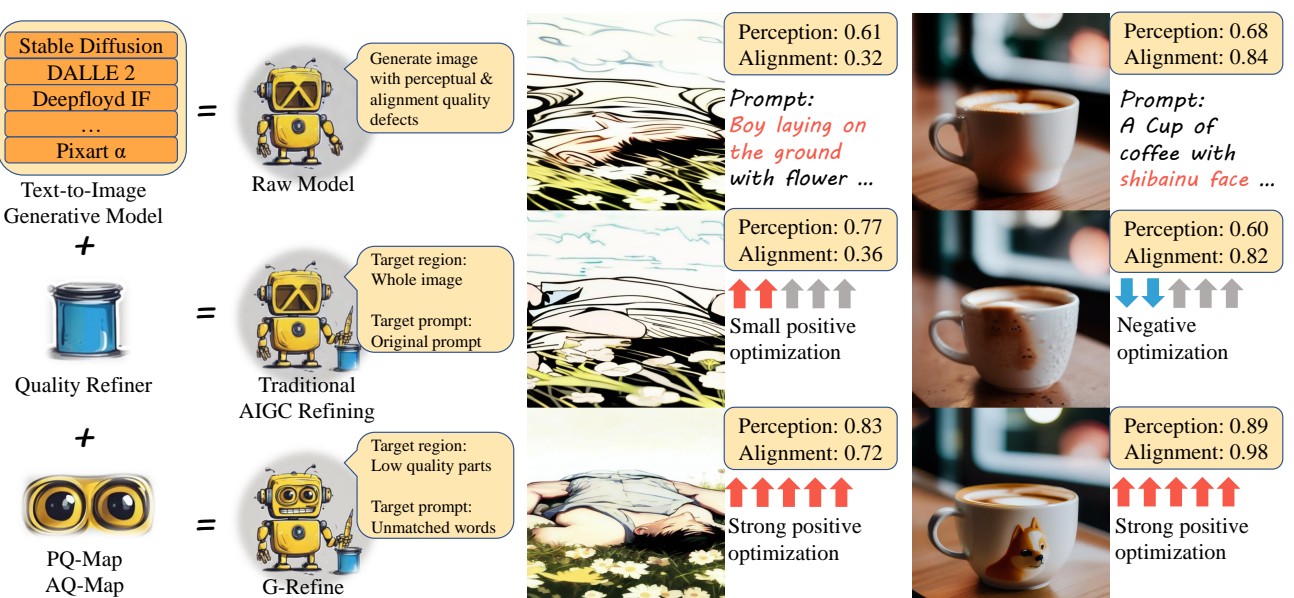

**Figure 1: The original AIGIs from AIGIQA-20K, optimized by different refiners in terms of perceptual and alignment quality. Inspired by quality indicators, G-Refine better optimizes low-quality regions while avoiding affecting the high-quality regions.**

## ABSTRACT

With the evolution of Text-to-Image (T2I) models, the quality defects of AI-Generated Images (AIGIs) pose a significant barrier to their widespread adoption. In terms of both perception and alignment, existing models cannot always guarantee high-quality results. To mitigate this limitation, we introduce G-Refine, a general image quality refiner designed to enhance low-quality images without compromising the integrity of high-quality ones. The model is composed of three interconnected modules: a perception quality indicator, an alignment quality indicator, and a general quality enhancement module. Based on the mechanisms of the Human Visual System (HVS) and syntax trees, the first two indicators can respectively identify the perception and alignment deficiencies, and the last module can apply targeted quality enhancement accordingly. Extensive experimentation reveals that when compared to alternative optimization methods, AIGIs after G-Refine outperform in 10+ quality metrics across 4 databases. This improvement significantly contributes to the practical application of contemporary T2I models, paving the way for their broader adoption.

## CCS CONCEPTS

• **Computing methodologies → Computer vision tasks**.

## KEYWORDS

AI-Generated Content, Image Quality Assessment, Text-to-Image Alignment, Image Restoration, Syntactic Parsing

## 1 INTRODUCTION

Text-to-Image (T2I) generation models have revolutionized the production and consumption of visual content. These models, guided by free-form text prompts, aim to generate high perceptual quality images that closely match the text. Recent advancements in diffusion models have led to significant leaps in T2I capabilities, making AI-generated images (AIGIs) increasingly relevant for advertising, entertainment, and even scientific research. However, the quality of AIGIs varies significantly, hindering their widespread adoption in industrial production. According to Hugging Face, over 10,000 T2I models have been developed since 2024. While some advanced models can address the challenge of high-quality generation, their usage is much lower than that of mainstream models like Stable Diffusion (SD) 1.5. Older models deployed by users often yield subpar results due to their earlier development. Additionally, even the latest models like Playground v2.5 suffer from inconsistent generation performance, with both master-class artworks and Low-Quality (LQ) AIGIs coexisting.

To ensure high-quality T2I generation in the industry, several solutions are employed. The most common approach is multiple runs

**Table 1: Using different optimizers for AI-Generated Images with low/high quality. [Keys: ✅✅ strong positive, ✅ positive, ⚫ zero, ❌ negative optimization.]**

| Optimizer | Input | LQ optimization | | HQ optimization | |
|---|---|---|---|---|---|
| | | Percept | Align | Percept | Align |
| Restoration | Image | ✅ | ⚫ | ✅ | ⚫ |
| Reconstruction | Image, text | ✅ | ✅ | ✅ | ✅ |
| Refinement | Image, text | ✅✅ | ✅✅ | ❌ | ❌ |
| Proposed | Image, text | ✅✅ | ✅✅ | ✅ | ✅ |

followed by manual selection, reporting only the best-quality AIGI. However, this method incurs significant computational waste and inevitable human labeling. More efficient methods include modifying T2I's U-Net to improve the perceptual quality or adjusting the text encoder for better alignment, such as FreeU [30], TextCraftor [17], and DPT [24]. However, these methods require access to the original model's parameters, making them only suitable for development rather than online consumption for the user end.

To overcome these challenges and facilitate the deployment of AIGI at the user end, a tailored optimization strategy is needed. This strategy should focus on generating images directly based on prompts, without relying on complex backend processes. However, there are three major obstacles: (**i**) Understanding the perception defects of AIGI. Unlike traditional Image Quality Assessment (IQA) methods that primarily address distortions such as blur and noise in Natural Sense Images (NSIs), AIGI's defects stem from hardware limitations and technical limitations, like unnaturalness and artifacts. To accurately assess AIGI quality, a new approach is required to distinguish these unique types of artifacts. Moreover, beyond a single quality score, a pixel-level ***perceptual quality map*** is needed to locate spatial quality defects. Limited by spatial relationship knowledge, it's difficult for existing IQA indicators to expand the overall score into a two-dimensional weight map. (**ii**) T2I alignment also requires similar maps instead of scores. The ***alignment quality map*** should indicate how well each part of the prompt corresponds to the generated image, and combine these insights into a comprehensive map. This task is more complex than simply assessing perceptual quality, as it involves understanding the semantic structure of the prompt. (**iii**) After identifying LQ regions, the challenge lies in optimizing them without compromising the quality of the rest of the image. ***Optimization balancing*** must be struck between applying just enough optimization to improve the image without introducing artifacts in High-Quality (HQ) regions. Thus, we propose G-Refine, a general quality optimizer as shown in Figure[1] 1 with the following contributions:

**PQ-Map**: An accurate perceptual quality map indicator. It can accurately understand the connotation of the word "quality", especially the quality defects of AIGIs. Considering the three quality-related factors (rationality, naturalness, and technical quality), it can accurately identify LQ regions for AIGIs. While outputting a 2D map, its performance can even ensemble single score models.

**AQ-Map**: An efficient alignment quality map indicator. By conducting syntactic parsing on a syntax tree, it can divide the prompt

---

into nodes representing different semantic information and analyze the relationship between the nodes. For nodes that do not align with the original AIGI, it uses the backtracking method to increase the weight of the ancestor node to give a complete alignment map.

**Balanced-refiner**: An optimization strategy for AIGI refiners. Inspired by PQ/AQ-Map, the refiner will **retain the HQ while improving LQ**. The model specifically consists of two stages. Stage 1 is similar to the traditional Refiner to fundamentally modify LQ; stage 2 refers to the restoration model by tuning LQ and HQ altogether. On 4 AIGI databases and 8 T2I generation models, compared to sota optimizers, G-Refiner has remarkable advantages in 9 perceptual quality and 4 alignment quality indicators.

## 2 RELATED WORKS

Without changing the internal generative model, to optimize AIGIs only through the prompt-image pairs, existing optimization strategies are mainly divided into the following categories as Table 1, which we summarize as three R's.

**Restoration**: Treat AIGIs directly as NSIs by leveraging Super Resolution (SR) or Image Restoration (IR) algorithms through Convolutional Neural Networks (CNNs) based on prior knowledge. This method can improve the perceptual quality, but it does not support text modality as input. As the prompt cannot be used as a reference, the alignment quality is almost unchanged.

**Reconstruction**: A text-guided IR technique for AIGIs using the CLIP[25] model to encode prompts. This approach modifies low-level image features referring to the prompt, such as adjusting global brightness or altering the colors of an object, thereby enhancing alignment to a certain extent. However, its effectiveness is limited when dealing with LQ images, as it cannot significantly alter object structures. Similarly, when the alignment quality is poor, the model fails to generate non-existent objects from the prompt. Consequently, the overall optimization impact of this strategy is insufficient across both image quality dimensions.

**Refinement**: According to the prompt, AIGIs can be significantly modified at the semantic level. Among them, the conservative Refine strategy will denoise the image at a lower intensity. This cascade paradigm (generation + refiner) has been widely used in today's T2I models, such as IF [7], SDXL [23], and SD Cascade [22]. A generator first provides a rough outline, then optimized through one or more refiners. A more radical strategy is to use the image directly as the starting point and perform the whole diffusion process. Compared with Reconstruction and Restoration, it can significantly optimize LQ regions. However, it usually contains certain AI artifacts, indicating an upper limit to its capabilities. While improving LQ, there will be negative optimization of HQ regions.

Therefore, distinguishing the LQ regions from the HQ is of great significance. However, though most of the existing IQA and T2I alignment indicators have excellent performance, their outputs are limited to a single score. Only Paq-2-Piq [42] supports the perceptual quality map and CLIP-Surgery [18] supports the alignment quality map. Unfortunately, the performance of these two methods is far inferior to the former, whose results are inconsistent with the subjective preference of the Human Visual System (HVS). Therefore, towards a targeted optimization of AIGIs, better quality map indicators are needed to inspire the refiner.

---

[1]The perceptual and alignment quality is from Q-Align [37] and CLIPScore [25]

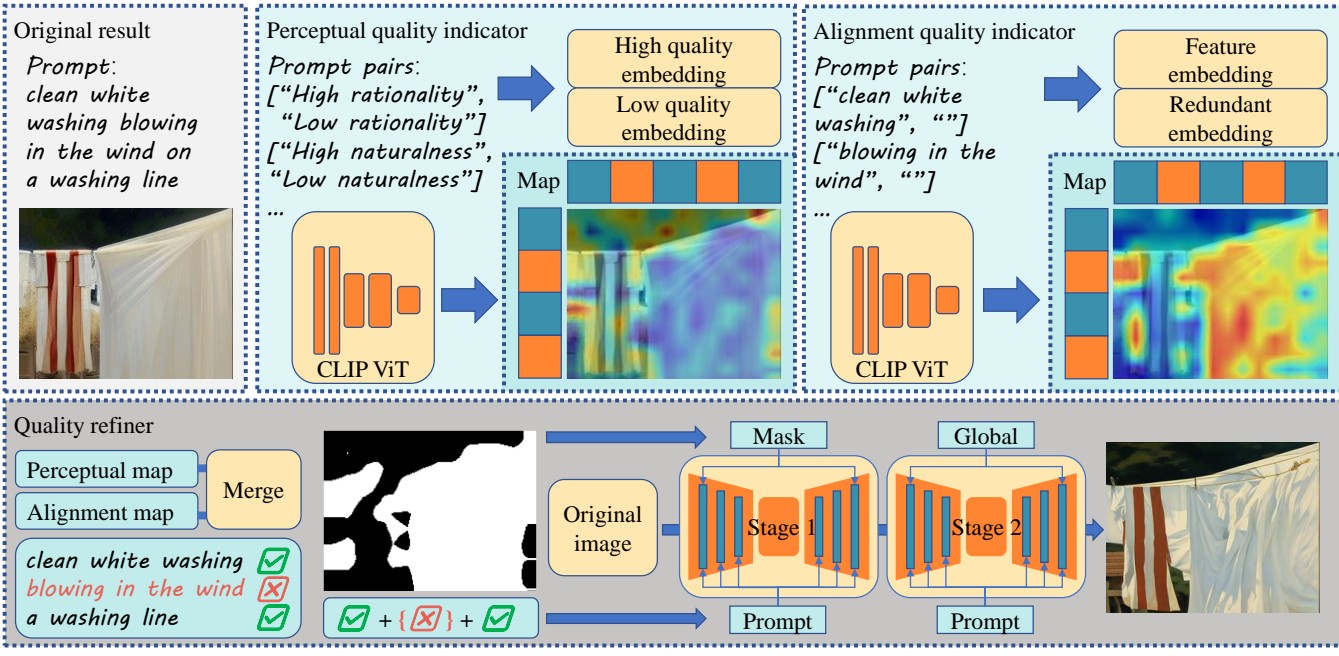

**Figure 2: Framework of G-Refine, including a perceptual quality and an alignment quality indicator module. The refining process is targeted at optimizing unmatched prompts and both maps. The perceptual quality is optimized by introducing more texture while the alignment quality is improved by implementing "blowing in the wind" into the image.**

## 3 PROPOSED METHOD

The framework of the proposed method is briefly illustrated in Figure 2, which includes a perceptual quality, an alignment quality indicator module for quality maps; and a quality refiner that merges quality maps spatially while emphasizing unmatched prompts semantically, towards a perceptual/alignment joint refinement.

### 3.1 Perceptual Quality Indicator

This PQ-Map module adjusts CLIP's image and text encoders respectively, thereby obtaining the perceptual quality weight map of the image. Intuitively, using CLIP to find the region correlated with the word 'high quality' is the most direct method. However, existing research [27] reveals that CLIP tends to prioritize background over foreground, which contradicts the HVS mechanism. Therefore, referring to the solution of CLIP-Surgery [18], PQ-Map first changes CLIP's original QKV self-attention into VVV:

$$C_I.\text{QKV} = \text{softmax}(V \cdot V^T \cdot \frac{1}{||V||_2}) \cdot V, \qquad (1)$$

where $V$ stands for value parameters for CLIP image encoder $C_I(\cdot)$ and $I$ represents the original AIGI. Next, we also modify the text encoder. Existing Segmentation Model [12] can easily identify objects such as 'cats' and 'dogs'. However, the 'perceptual quality' is different from the objective concept, which is a highly subjective concept that combines multiple factors. Therefore, the text encoder should not take "perceptual quality" directly, but decompose it into quality-related factors and encode them together. According to subjective analysis [4], AIGIs perceptual quality defects mainly include three categories: technical, rationality, and naturalness. On this

basis, a $4 \times 2$ token embedding with 512 length $T_p$ is given:

$$T_p = C_T(t_0, t_1, t_2, t_3), \qquad (2)$$

where the text encoder $C_I(\cdot)$ process the text pairs of CLIPIQA [33] $t_0$ representing the overall perceptual quality, and text pairs $t_{1\sim3}$ for three perceptual quality defects. Generally, the perceptual quality follows the cask effect. The excellence of a single factor cannot guarantee a score improvement, but its defects will inevitably lead to a decrease. Thus we express the perceptual quality map $P$ and the score $p$ as:

$$\begin{cases} L_{raw} = C_I \odot (T_p[:,0] - T_p[:,1]) \\ L_{per} = L_{raw}[0] \cdot \prod_{i=1}^{3} \min(\frac{L_{raw}[i]}{\alpha[i]}, 1) \\ P = \text{BIC}(L_{per}), p = L_{per}[0], \end{cases} \qquad (3)$$

where $L_{(raw,per)}$ stands for the raw logit embedding, and final perceptual quality embedding combined with four logits. $\text{BIC}(\cdot)$ rescales the logit into the size of $I$ from bi-cubic interpolation. Logits below $\alpha$ will introduce a penalty to $L_{per}$. From the difference between $p$ and subjective quality annotation, PQ-Map can update $T_p$ to a better embedding, in order words, to fully interpret the complex connotation of the word 'perceptual quality', as shown in Figure 3.

In the subsequent quality map calculation, the token embedding layer can be disabled, without extracting any text features, but directly input the features $T_p$ representing the perceptual quality into the text encoder. Figure 4 shows the map results using the original CLIP or the improved encoder. The map obtained from the original image encoder has almost no regularity and only shows high correlation at a few meaningless points; after improving the image

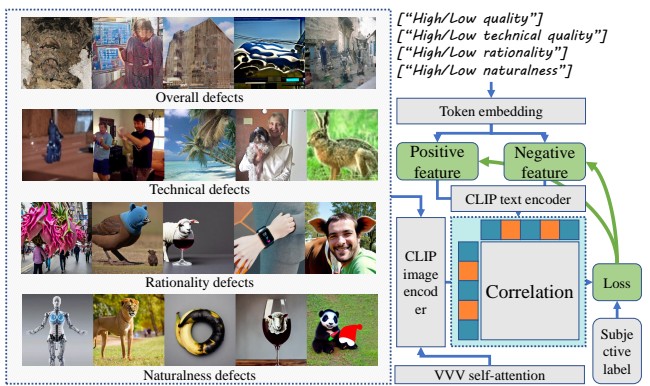

Figure 3: Using overall perceptual quality, and (technical, rational, natural) defected images to train the CLIP model. Both image and text encoder are modified in terms of self-attention and token embedding.

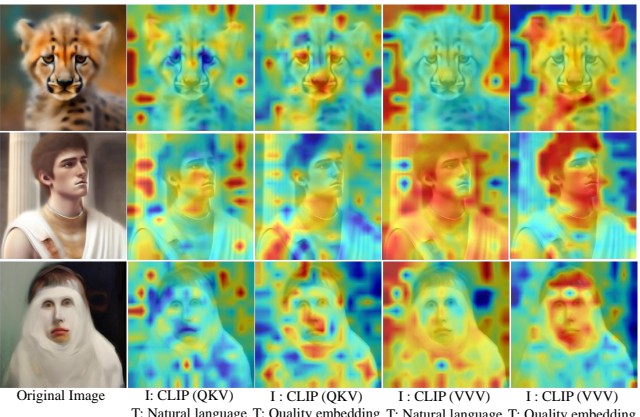

Figure 4: Visualization result of the perceptual quality map, using the original CLIP or improved image/text encoders. The original image encoder generates meaningless results while the original text encoder labels reversely. Reasonable results are available only when two encoders are modified.

encoder, the original text encoder can perform certain semantic segmentation of the image, but due to the limited understanding of 'perceptual quality', it cannot mark all LQ regions and might even reverse LQ and HQ. An accurate perceptual quality map is available only by applying two improved encoders simultaneously. The results above prove the modifications of the two encoders bring significant positive effects jointly to the perceptual quality map.

## 3.2 Alignment Quality Indicator

This AQ-Map module first decomposes the prompt into phrases with different semantic meanings, analyzes the T2I alignment of each phrase, and finally merges them to obtain an alignment quality map for the whole prompt. Considering the length of prompts and complex subordination relationships between words, it is unrealistic to directly input prompts into $C_T$ to calculate alignment. Therefore,

---

**Algorithm 1** get_phrase_ancestor

1: **function** GET_PHRASE_ANCESTOR($pns, phs, Tree, obj$)
2:     $ans, stack \leftarrow \{\}, \{\}$
3:     **for** $pn$ in $pns$ **do**
4:         $ans[pn] \leftarrow pn$
5:         **for** $ph$ in $phs$ **do**
6:             $stack[ph] \leftarrow pn$
7:     $pq \leftarrow [Tree.root]$
8:     **while** $pq$ is not empty **do**
9:         $obj \leftarrow pq.head$
10:         **for** $child$ in $obj.children$ **do**
11:             $pq.add(child)$
12:             **if** $stack[child.ancestor] \neq stack[child]$ **then**
13:                 **find the first ancestor with tag "NOUN"**
14:                 **for** $ancestor$ in $child.ancestor$ **do**
15:                     **if** $ancestor.pos == "NOUN"$ **then**
16:                         $ans[stack[child]] \leftarrow stack[ancestor]$
17:                     **break**
18:         $pq.dequeue$
19:     **return** $ans$

---

we build a syntax tree $Tree$ from the NLTK($\cdot$) package, using nouns $pns$ as the clustering center, and assign each tree node to different phrases $phs$:

$$\begin{cases} Tree = \text{NLTK}(prompt) \\ phs = pns = Tree.Noun \\ phs[j].append(Tree[||pns[j], Tree||_{\min}]), \end{cases} \quad (4)$$

where $prompt$ is the original input prompt. The $phs$ is initialized as noun nodes $pns$, while the remaining nodes will be associated with the closest $pns$ to form several different phrases. The above method effectively segments prompts, at the cost of destroying the syntax tree. To solve this problem, according to the original dependency relationship of $pns$, AQ-Map allocates a new ancestor $ans$ for each $phs$ using the Algorithm 1. After segmentation, AQ-Map calculates the alignment quality map and score ($A_{phs}, a_{phs}$) for each phrase separately:

$$\begin{cases} A_{phs}[j, :] = \text{softmax}(C_I(I) \odot C_T(phs[j], \text{""})) \\ a_{phs}[j] = A_{phs}[j, 0], \end{cases} \quad (5)$$

where index $j \in [0, \text{len}(phs)-1]$. $pns$ experience a similar computation with ($A_{pns}, a_{pns}$). An empty string "" is encoded as redundant feature and removed by softmax($\cdot$). Therefore, AQ-Map can summarize the alignment defect into the following two types, with typical examples shown in Figure 5:

- Noun unmatched: $a_{pns} < a_{bound}$ which indicates the noun doesn't exist in the image. Here, the whole phrase should be drawn on the correlated region of its ancestor node.
- Adj. unmatched: $a_{phs} < a_{pns}$ which means the noun exists, but adjectives are not well-represented on it. In this case, the phrase should be drawn on the region itself.

Thus, initialize the overall alignment quality map $A = 1$, AQ-Map implement the weight of $A_{pns}$ on $A$ as:

$$A = \begin{cases} A \cdot A_{pns}[j] & a_{pns} < a_{bound} \\ A \cdot A_{pns}[phs[j].ancestor] & a_{phs} < a_{pns}. \end{cases} \quad (6)$$

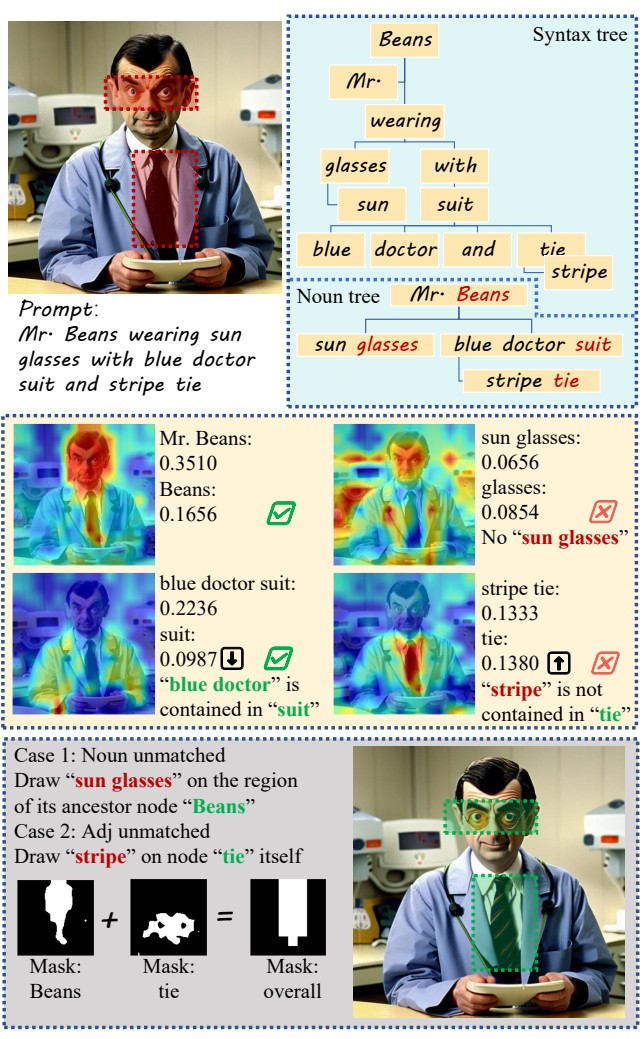

**Figure 5: The mechanism of identifying alignment quality defects. Include syntax tree construction, quality defect identification, and mask processing. Both unmatched nouns and adjectives can be enhanced on their correlated region.**

For all index $j$ with two alignment defects above, the phrases are also emphasized in the prompt for stronger refining strength:

$$prompt = prompt + 0.1 phs[j]. \tag{7}$$

Then following the cask effect same as equation (3), the alignment quality score $a$ will be:

$$a = C_I(I) \odot C_T(prompt) \prod_j \min(\frac{a_{phs}[j]}{\beta}, 1), \tag{8}$$

where for each phrase, alignment score below $\beta$ will introduce penalty to $a$. From this, AQ-Map emphasized the phrases unaligned with the original image and mapped their corresponding regions that needed improvement based on the alignment defects.

## 3.3 Quality Refiner

The quality refiner is designed to optimize the image for both perceptual and alignment quality. This model is cascaded by a refinement and a restoration stages, the former conducts a strong denoising process according to the map and score given above, and the latter performs a mild denoising globally. Since the targeted refining region has a relatively lower quality and more alignment defects, the probability densities for two stages are $z_1/z_2$:

$$\begin{cases} z_1 = \frac{p+a}{2} \cdot \text{QKV}(prompt, \text{Bi}(1 - P + A)) \\ z_2 = \delta \cdot \text{QKV}(prompt, 1), \end{cases} \tag{9}$$

where the strength of $z_1$ depends on quality $(p, a)$ while $z_2$ takes a extremely small strength $\delta$. $\text{Bi}(\cdot)$ binarizes the map into a mask. After obtaining the probability densities, we can denoise the image with the refined result $R$:

$$R = \mathcal{D}_{n_2}^{z_2} \cdots \mathcal{D}_1^{z_2}(\mathcal{D}_{n_1}^{z_1} \cdots \mathcal{D}_1^{z_1}(I)), \tag{10}$$

where $\mathcal{D}_n^z$ denotes the diffusion operation at the $n$-th iteration and $(n_1, n_2)$ are the specific diffusion steps for each stage.

## 4 EXPERIMENT

### 4.1 Validation Databases

To assess the efficacy of the proposed G-Refine method across diverse generative models, we conducted performance evaluations on four commonly used AIGI databases: DiffusionDB [35], GenImage [47], AGIQA-1K [46], and AGIQA-3K [16]. Considering the huge scale of the first two, we randomly selected 3,000 images for refining, while using the complete set for the latter two. Since these AIGIs only come from traditional models like SD1.5, to verify our versatility for other generative models, we randomly generated 500 images each by 7 commonly-used models [2] for our G-Refine pipeline to optimize. Besides the whole G-Refine, we adopt the subjective scoring result from the two most popular AIGI quality databases, AIGIQA-20K (testing set) [14] and AGIQA-3K (full set) [16], to validate the effectiveness of PQ/AQ-Map quality indicators. These databases contain fine-grained Mean Opinion Score (MOS) as perceptual and alignment quality labels, to measure the correlation between them and objective evaluation results.

### 4.2 Experiment Settings

For quality optimization, we include 13 representative methods in different categories as baselines, including (**Restoration**): RFDN [20], Swin2SR [5], StableSR [34], and DASR [36]; (**Reconstruction**): SD-Upscale [28], Instructpix2pix [1], DiffBIR [19], PASD [41]; and (**Refinement**): SDXL-Refiner [23], SD (full-model) [28], SDXL (full-model) [29], InstructIR [6], and Q-refine [15]. All models are run by 20 iterations for a fair comparison. The optimization quality is comprehensively evaluated in 13 indicators, namely four (**Perceptual**)[3]: CLIPIQA [33], UNIQUE [44], LIQE [45], DBCNN [43], TOPIQ [2], CNNIQA [10], MUSIQ [11], BRISQUE [21], Q-Align [37]; and nine (**Alignment**): CLIPScore [25], ImageReward [40], PicScore [13], and HPSv2 [38]. Effective models should have lower BRISQUE and higher scores for other quality indicators. For quality assess-

---

[2]The 7 models are selected by the download times on huggingface. For some other advanced models with less popularity, the refining result is attached in the supplementary.
[3]FID is not considered as it shows less correspondence with human preference.

**Table 2: Using different quality optimizers on GenImage database. Abbreviations: BRISQ: BRISQUE; CLIPS: CLIPScore; ImgRw: ImageReward; PicS: PicScore. Left/right for perceptual/alignment quality. [Key: Best; Second Best; Negative optimization ].**

| Optimizer | Method / Indicator | CLIPIQA↑ | UNIQUE↑ | LIQE↑ | DBCNN↑ | TOPIQ↑ | CNNIQA↑ | MUSIQ↑ | BRISQ↓ | Q-Align↑ | CLIPS↑ | ImgRw↑ | PicS↑ | HPSv2↑ |
|---|---|---|---|---|---|---|---|---|---|---|---|---|---|---|
| N/A | *Original Images* | 0.6911 | 1.1634 | 3.6417 | 0.6004 | 0.5605 | 0.6347 | 66.337 | 17.344 | 3.9609 | 0.9809 | 0.0724 | 0.7117 | 0.2563 |
| Restoration | RFDN (ECCV2020) | 0.6558 | 1.1119 | 3.5193 | 0.5579 | 0.5176 | 0.6225 | 65.416 | 22.748 | 3.8633 | 0.9830 | 0.0479 | 0.7027 | 0.2556 |
| | Swin2SR (ECCV2022) | 0.6982 | 1.1651 | 3.6435 | 0.6021 | 0.5617 | 0.6346 | 66.378 | 17.379 | 3.9609 | 0.9807 | 0.0721 | 0.7124 | 0.2563 |
| | StableSR (Arxiv2023) | 0.7306 | 1.3611 | 3.9148 | 0.6639 | 0.6441 | 0.6497 | 71.360 | 14.595 | 3.9883 | 0.9807 | -0.0577 | 0.6858 | 0.2574 |
| | DASR (CVPR2021) | 0.6345 | 0.8340 | 2.8897 | 0.6309 | 0.5205 | 0.6921 | 61.111 | 53.082 | 3.4648 | 0.9784 | -0.0349 | 0.6498 | 0.2533 |
| Reconstruct-ion | SD-Upscale (CVPR2022) | 0.6655 | 1.1363 | 3.5722 | 0.5335 | 0.5178 | 0.5998 | 66.119 | 24.979 | 3.8574 | 0.9835 | 0.0373 | 0.6993 | 0.2558 |
| | Instructpix (CVPR2023) | 0.6204 | 0.8441 | 3.0683 | 0.5098 | 0.4791 | 0.5588 | 61.714 | 26.031 | 3.7422 | 0.9814 | 0.0255 | 0.6701 | 0.2562 |
| | DiffBIR (Arxiv2024) | 0.7313 | 1.1327 | 3.6557 | 0.6113 | 0.5853 | 0.6436 | 65.987 | 11.505 | 3.9297 | 0.9843 | -0.1569 | 0.6769 | 0.2542 |
| | PASD (Arxiv2024) | 0.7214 | 1.3444 | 4.0707 | 0.6534 | 0.6501 | 0.6280 | 69.691 | 13.852 | 4.0156 | 0.9845 | 0.0038 | 0.7358 | 0.2565 |
| Refinement | SDXL-Refiner (ICLR2023) | 0.6068 | 0.9338 | 3.2709 | 0.4932 | 0.4723 | 0.5062 | 62.343 | 30.049 | 3.8672 | 0.9841 | 0.2027 | 0.7153 | 0.2571 |
| | SD (CVPR2022) | 0.6110 | 0.8163 | 2.9576 | 0.4903 | 0.4636 | 0.4212 | 60.341 | 24.632 | 3.6191 | 0.9335 | 0.0013 | 0.6591 | 0.2545 |
| | SDXL (ICLR2024) | 0.6390 | 0.9696 | 3.3239 | 0.5074 | 0.4968 | 0.5846 | 63.598 | 28.300 | 3.9355 | 0.9918 | 0.2683 | 0.7652 | 0.2578 |
| | InstructIR (Arxiv2024) | 0.6866 | 1.0776 | 3.5751 | 0.5819 | 0.5566 | 0.6371 | 64.668 | 28.945 | 3.8867 | 0.9871 | -0.0395 | 0.6680 | 0.2542 |
| | Q-Refine (ICME2024) | 0.7358 | 1.1833 | 3.7128 | 0.6122 | 0.5877 | 0.6491 | 66.943 | 11.630 | 3.9668 | 0.9889 | -0.1109 | 0.6818 | 0.2548 |
| **G-Refine (Ours)** | | 0.7444 | 1.5139 | 4.1280 | 0.6817 | 0.6679 | 0.6603 | 73.004 | 11.225 | 4.3366 | 0.9906 | 0.2290 | 0.7656 | 0.2604 |

**Table 3: Using different quality optimizers on DiffusionDB database. Abbreviation and keys follow Table 2.**

| Optimizer | Method / Indicator | CLIPIQA↑ | UNIQUE↑ | LIQE↑ | DBCNN↑ | TOPIQ↑ | CNNIQA↑ | MUSIQ↑ | BRISQ↓ | Q-Align↑ | CLIPS↑ | ImgRw↑ | PicS↑ | HPSv2↑ |
|---|---|---|---|---|---|---|---|---|---|---|---|---|---|---|
| N/A | *Original Images* | 0.7147 | 0.9106 | 3.5127 | 0.6094 | 0.5714 | 0.6263 | 65.003 | 15.093 | 3.8672 | 0.9822 | -0.1085 | 0.7372 | 0.2550 |
| Restoration | RFDN (ECCV2020) | 0.7147 | 0.9117 | 3.5127 | 0.6093 | 0.5714 | 0.6256 | 65.004 | 15.076 | 3.8672 | 0.9822 | -0.1089 | 0.7372 | 0.2550 |
| | Swin2SR (ECCV2022) | 0.7125 | 0.9122 | 3.5156 | 0.6103 | 0.5725 | 0.6265 | 65.000 | 15.093 | 3.8711 | 0.9824 | -0.1089 | 0.7371 | 0.2550 |
| | StableSR (Arxiv2023) | 0.6920 | 1.1946 | 3.5930 | 0.6458 | 0.6045 | 0.6301 | 69.200 | 16.035 | 3.8789 | 0.9637 | -0.2011 | 0.6592 | 0.2551 |
| | DASR (CVPR2021) | 0.6411 | 0.4695 | 2.8614 | 0.6526 | 0.5802 | 0.6811 | 61.413 | 15.539 | 3.4785 | 0.9781 | -0.1997 | 0.7016 | 0.2552 |
| Reconstruct-ion | SD-Upscale (CVPR2022) | 0.6576 | 0.9454 | 3.4477 | 0.5756 | 0.5459 | 0.5837 | 65.256 | 14.392 | 3.8340 | 0.9724 | -0.1393 | 0.7302 | 0.2545 |
| | Instructpix (CVPR2023) | 0.6325 | 0.6157 | 3.2975 | 0.5007 | 0.4739 | 0.5056 | 59.299 | 16.236 | 3.6270 | 0.9690 | -0.1068 | 0.7001 | 0.2542 |
| | DiffBIR (Arxiv2024) | 0.7111 | 0.9046 | 3.2790 | 0.5691 | 0.5238 | 0.6265 | 67.302 | 11.438 | 3.8152 | 0.9784 | -0.1835 | 0.7057 | 0.2547 |
| | PASD (Arxiv2024) | 0.7047 | 1.3098 | 3.7973 | 0.6393 | 0.6568 | 0.6436 | 67.146 | 16.468 | 3.8215 | 0.9819 | -0.1836 | 0.6721 | 0.2517 |
| Refinement | SDXL-Refiner (ICLR2023) | 0.5602 | 0.4098 | 2.8306 | 0.4973 | 0.4426 | 0.4613 | 61.377 | 14.429 | 3.5816 | 0.9686 | 0.0380 | 0.6948 | 0.2517 |
| | SD (CVPR2022) | 0.6553 | 0.6896 | 2.7884 | 0.4896 | 0.4611 | 0.4816 | 59.392 | 21.925 | 3.5996 | 0.9438 | -0.0269 | 0.6961 | 0.2549 |
| | SDXL (ICLR2024) | 0.6203 | 0.7874 | 3.7208 | 0.5772 | 0.5501 | 0.5899 | 63.440 | 15.161 | 3.9785 | 0.9895 | 0.1402 | 0.7861 | 0.2575 |
| | InstructIR (Arxiv2024) | 0.7370 | 0.9459 | 3.6083 | 0.6402 | 0.6181 | 0.6410 | 63.179 | 15.192 | 3.9414 | 0.9863 | -0.1738 | 0.7111 | 0.2529 |
| | Q-Refine (ICME2024) | 0.7194 | 1.0040 | 3.4199 | 0.5981 | 0.5593 | 0.6431 | 66.085 | 12.018 | 3.9336 | 0.9915 | -0.1834 | 0.6728 | 0.2533 |
| **G-Refine (Ours)** | | 0.7153 | 1.4706 | 3.8922 | 0.6762 | 0.6471 | 0.6569 | 72.193 | 13.934 | 4.2034 | 0.9933 | 0.1412 | 0.7277 | 0.2593 |

ment, we apply 12 advanced quality indicators for comparison, as (**Perceptual**) DBCNN [43], CLIPIQA [33], CNNIQA [10], HyperIQA [31], NIMA [32], and Paq2Piq [42]; (**Alignment**) CLIPScore [25], ImageReward [40], HPSv1 [39], HPSv2 [38], and CLIP-Surgery [18]. We measure the correlation between subjective labeling objective prediction, namely Spearman Rank-order Correlation Coefficient (SRCC) and Pearson Linear Correlation Coefficient (PLCC). Higher SRCC/PLCC indicates better prediction monotonicity/accuracy. All quality indicators are fine-tuned on the AIGIQA-20K (training set). During the training process of PQ-Map, we froze the parameters of the image encoder as CLIP-Surgery [18] and only updated the text encoder. While for the AQ-Map, the parameters of the image encoder are initialized as ImageReward [40]. The refiner Stage 1 adapts SDXL-Inpainting [29] model mixing PQ/AQ-Map as a mask; Stage 2 applies PASD [41] model globally. Each stage takes half of the iterations. We generate original AIGIs and train quality indicators for 50 epochs using Adam optimizer on a server with four NVIDIA RTX A6000, and validate the quality optimization/assessment performance on a local NVIDIA GeForce RTX 4090.

## 4.3 Quality Optimization Results

Table 2, 3, 4, and 5 listed the perceptual/alignment quality optimization result. G-Refine's advantages are primarily showcased in

its **superior positive optimization** capabilities for AIGI quality. Across 13 indicators assessed on 4 databases, G-Refine secured first or second place in over 90% of the cases (47/52). The performance is more exceptional on the standard AIGI database GenImage, and the AGIQA-3K with significant internal quality variation. Though StableSR and SDXL also exhibit certain optimization for perceptual and alignment quality, G-Refine stands out by offering general optimization for both qualities. G-Refine's ability to excel in BRISQUE and LIQE, which represent signal fidelity and aesthetics, respectively, underscores its multi-dimensional perceptual quality optimization. Similarly, its dominance in CLIP and ImagReward, indicative of word-level and sentence-level semantic alignment, demonstrates its capacity to understand and enhance word relationships for images. Considering some indicators are inconsistent with human real preferences, to enhance the credibility of real scenarios, we considered the indicators most relevant to human subjective preferences, namely Q-Align and HPSv2. Notably, G-Refine achieved the best in both, indicating its ability to enhance human genuine contentment with AIGIs beyond fixed indicators.

Another key strength of G-Refine lies in its **minimized negative optimization**. On 52 indexes, we marked results with lower quality than the original image. It can be seen that almost all other methods have experienced more than 10 negative optimizations

**Table 4: Using different quality optimizer on AGIQA-1K database. Abbreviation and keys follow Table 2.**

| Optimizer | Method / Indicator | CLIPIQA↑ | UNIQUE↑ | LIQE↑ | DBCNN↑ | TOPIQ↑ | CNNIQA↑ | MUSIQ↑ | BRISQ↓ | Q-Align↑ | CLIPS↑ | ImgRw↑ | PicS↑ | HPSv2↑ |
|---|---|---|---|---|---|---|---|---|---|---|---|---|---|---|
| N/A | *Original Images* | 0.6314 | 1.2237 | 3.6079 | 0.5730 | 0.5386 | 0.6332 | 68.330 | 31.727 | 3.5527 | 0.6648 | -1.4530 | 0.4079 | 0.2465 |
| Restoration | RFDN (ECCV2020) | 0.6324 | 1.2249 | 3.6054 | 0.5732 | 0.6338 | 0.6383 | 68.304 | 31.754 | 3.5566 | 0.6617 | -1.4509 | 0.4077 | 0.2464 |
| | Swin2SR (ECCV2022) | 0.6374 | 1.2250 | 3.6076 | 0.5746 | 0.6397 | 0.6285 | 68.365 | 31.515 | 3.5566 | 0.6659 | -1.4522 | 0.4083 | 0.2465 |
| | stablesr (Arxiv2023) | 0.7664 | 1.5365 | 4.4375 | 0.7128 | 0.7233 | 0.7597 | 75.111 | 14.401 | 3.9629 | 0.7025 | -1.4702 | 0.4435 | 0.2489 |
| | DASR (CVPR2021) | 0.6337 | 1.0618 | 3.3594 | 0.6271 | 0.5470 | 0.6933 | 62.765 | 45.635 | 3.3086 | 0.6663 | -1.4715 | 0.3990 | 0.2464 |
| Reconstruct-ion | SD-Upscale (CVPR2022) | 0.6262 | 1.2746 | 3.7111 | 0.5578 | 0.5477 | 0.6236 | 69.246 | 36.646 | 3.5488 | 0.6694 | -1.4621 | 0.4086 | 0.2463 |
| | Instructpix (CVPR2023) | 0.6306 | 1.1273 | 3.4683 | 0.5646 | 0.5261 | 0.6330 | 75.321 | 31.365 | 3.4824 | 0.6564 | -1.4671 | 0.4031 | 0.2465 |
| | DiffBIR (Arxiv2024) | 0.6994 | 1.8592 | 4.3359 | 0.6539 | 0.6482 | 0.6855 | 77.562 | 27.518 | 3.9258 | 0.6905 | -1.5073 | 0.4389 | 0.2455 |
| | PASD (Arxiv2024) | 0.6926 | 1.5653 | 4.3969 | 0.6829 | 0.6875 | 0.6532 | 73.684 | 23.431 | 3.9453 | 0.6899 | -1.5007 | 0.4230 | 0.2445 |
| Refinement | SDXL-Refiner (ICLR2023) | 0.7021 | 1.5065 | 4.3324 | 0.6185 | 0.5887 | 0.6550 | 76.521 | 52.893 | 3.9473 | 0.7841 | -1.1106 | 0.5319 | 0.2523 |
| | SD (CVPR2022) | 0.6523 | 1.3299 | 3.7332 | 0.5827 | 0.5520 | 0.5702 | 69.585 | 26.675 | 3.5508 | 0.9366 | -0.5347 | 0.5967 | 0.2561 |
| | SDXL (ICLR2024) | 0.5849 | 1.1592 | 3.6301 | 0.5251 | 0.5106 | 0.6161 | 68.755 | 36.184 | 3.6621 | 0.8261 | -0.8687 | 0.5738 | 0.2527 |
| | InstructIR (Arxiv2024) | 0.6766 | 1.4906 | 4.2460 | 0.6287 | 0.6135 | 0.6810 | 72.514 | 33.462 | 3.8398 | 0.9703 | -0.1914 | 0.6463 | 0.2584 |
| | Q-Refine (ICME2024) | 0.7394 | 1.6307 | 4.4129 | 0.6613 | 0.6486 | 0.6897 | 73.164 | 19.420 | 3.9785 | 0.9122 | -1.3007 | 0.4563 | 0.2475 |
| **G-Refine (Ours)** | | 0.7741 | 1.6773 | 4.6922 | 0.7351 | 0.7542 | 0.6594 | 76.487 | 9.3422 | 4.1703 | 0.9610 | -0.3064 | 0.6704 | 0.2611 |

**Table 5: Using different quality optimizer on AGIQA-3K database. Abbreviation and keys follow Table 2.**

| Optimizer | Method / Indicator | CLIPIQA↑ | UNIQUE↑ | LIQE↑ | DBCNN↑ | TOPIQ↑ | CNNIQA↑ | MUSIQ↑ | BRISQ↓ | Q-Align↑ | CLIPS↑ | ImgRw↑ | PicS↑ | HPSv2↑ |
|---|---|---|---|---|---|---|---|---|---|---|---|---|---|---|
| N/A | *Original Images* | 0.5941 | 0.9001 | 3.3994 | 0.5330 | 0.5187 | 0.5856 | 60.740 | 35.261 | 3.7500 | 0.9527 | -0.0727 | 0.6849 | 0.2508 |
| Restoration | RFDN (ECCV2020) | 0.5929 | 0.8986 | 3.3971 | 0.5325 | 0.5817 | 0.5867 | 60.713 | 35.197 | 3.7520 | 0.9521 | -0.0716 | 0.6849 | 0.2507 |
| | Swin2SR (ECCV2022) | 0.5996 | 0.9010 | 3.3997 | 0.5344 | 0.5195 | 0.5857 | 60.725 | 35.167 | 3.7559 | 0.9529 | -0.0710 | 0.6852 | 0.2465 |
| | StableSR (Arxiv2023) | 0.7453 | 1.3792 | 4.1906 | 0.6849 | 0.6805 | 0.7090 | 70.516 | 11.496 | 4.2539 | 0.9555 | -0.1249 | 0.7030 | 0.2539 |
| | DASR (CVPR2021) | 0.5302 | 0.5361 | 2.8626 | 0.5611 | 0.5079 | 0.6993 | 57.476 | 53.068 | 3.0566 | 0.9510 | -0.1927 | 0.6470 | 0.2490 |
| Reconstruct-ion | SD-Upscale (CVPR2022) | 0.5884 | 0.9373 | 3.4210 | 0.5196 | 0.5188 | 0.5977 | 62.294 | 35.139 | 3.8809 | 0.9449 | -0.1204 | 0.6771 | 0.2501 |
| | Instructpix (CVPR2023) | 0.5876 | 0.8473 | 3.3395 | 0.5212 | 0.5047 | 0.5958 | 69.330 | 35.132 | 3.6680 | 0.9448 | -0.1179 | 0.6705 | 0.2510 |
| | DiffBIR (Arxiv2024) | 0.6734 | 1.1664 | 3.7956 | 0.6358 | 0.6536 | 0.6815 | 67.950 | 15.634 | 4.1094 | 0.9608 | -0.1979 | 0.6796 | 0.2514 |
| | PASD (Arxiv2024) | 0.7161 | 1.8243 | 3.9960 | 0.6258 | 0.6372 | 0.6535 | 65.630 | 24.230 | 4.1222 | 0.9673 | -0.1674 | 0.7067 | 0.2521 |
| Refinement | SDXL-Refiner (ICLR2023) | 0.6694 | 1.2785 | 3.9161 | 0.5497 | 0.5372 | 0.6527 | 66.365 | 28.948 | 3.4399 | 0.9598 | 0.1350 | 0.7597 | 0.2523 |
| | SD (CVPR2022) | 0.6267 | 1.0776 | 3.6229 | 0.5299 | 0.5147 | 0.5702 | 64.414 | 30.296 | 3.7109 | 0.9595 | 0.1083 | 0.7187 | 0.2538 |
| | SDXL (ICLR2023) | 0.5358 | 0.7406 | 3.1452 | 0.4486 | 0.4610 | 0.5310 | 59.397 | 41.067 | 3.8652 | 0.9749 | 0.2310 | 0.7697 | 0.2527 |
| | InstructIR (Arxiv2024) | 0.6526 | 1.0183 | 3.6662 | 0.5666 | 0.5587 | 0.6207 | 63.344 | 39.948 | 3.8691 | 0.9934 | 0.0311 | 0.7009 | 0.2532 |
| | Q-Refine (ICME2024) | 0.7183 | 1.1291 | 3.7658 | 0.5990 | 0.5735 | 0.6539 | 89.897 | 22.001 | 4.1367 | 0.9783 | -0.0992 | 0.7027 | 0.2521 |
| **G-Refine (Ours)** | | 0.7717 | 1.5990 | 4.5163 | 0.7099 | 0.7168 | 0.6891 | 73.551 | 8.0697 | 4.3198 | 0.9865 | 0.2663 | 0.7643 | 0.2589 |

(only Q-Refine has less negative optimization but at the cost of limited positive optimization). In contrast, G-Refine produced only one negative optimization. This superiority stems from G-Refine's superior control over optimization intensity. Traditional methods often struggle to strike a balance, as enhancing LQ inevitably leads to degradation in HQ regions. However, G-Refine demonstrates a unique capability, achieving just one negative optimization. This demonstrates its capacity to discern between LQ and HQ regions, performing targeted, moderate denoising of defective areas without resorting to global operations.

Considering the above four databases are all generated by traditional, single T2I models, Figure 6 shows the performance[4] of the above optimizers on a variety of advanced T2I models, containing AnimateDiff [9] , DALLE2 [26], Dreamlike [8], IF [7], PixArt [3], SD1.5 [28], SD Cascade [22], and SDXL [29]. Intriguingly, G-Refine exhibits a stronger impact on models with lower initial quality, with notable collaborative optimization of perceptual/alignment quality for AnimateDiff, Dreamlike, SD1.5, and SD Cascade. For models with higher original generation quality, G-Refine still leads in perceptual quality, but the alignment optimization is less pronounced, particularly for PixArt, where all optimizers, including G-Refine, negatively affect alignment. Consequently, given the success of

---

[4]To simplify the image structure, we selected the three best-performing optimizers in Table 2-5 along with the original image for comparison.

G-Refine with traditional models, exploring further optimization of advanced models' generative quality (especially for alignment) is a pertinent research question.

## 4.4 Quality Assessment Results

The two sub-modules of G-Refine, namely PQ/AQ-Map, can also be used independently for quality evaluation tasks. Tables 6 and 7 illustrate their performance on AIGI-20K and cross-validated with AGIQA-3K. The existing methods generally excel in providing accurate quality scores, but are unable to offer quality maps. On the other hand, methods that support quality maps as outputs often have unacceptable correlations with human subjective ratings, rendering them less practical. PQ-Map and AQ-Map stand out in this regard, as they not only offer accurate scores but also produce quality maps that are both usable and comparable to the most advanced models in terms of perception and alignment quality evaluation. Their ability to output quality maps makes them highly applicable in tasks such as image annotation and restoration, with G-Refine serving as a prime example. We are eager to see the potential for these maps to be further integrated into related fields.

## 4.5 Ablation Study

To assess the individual impact of the two stages in G-Refine and the guidance provided by PQ/AQ-Map, we temporarily disabled stage

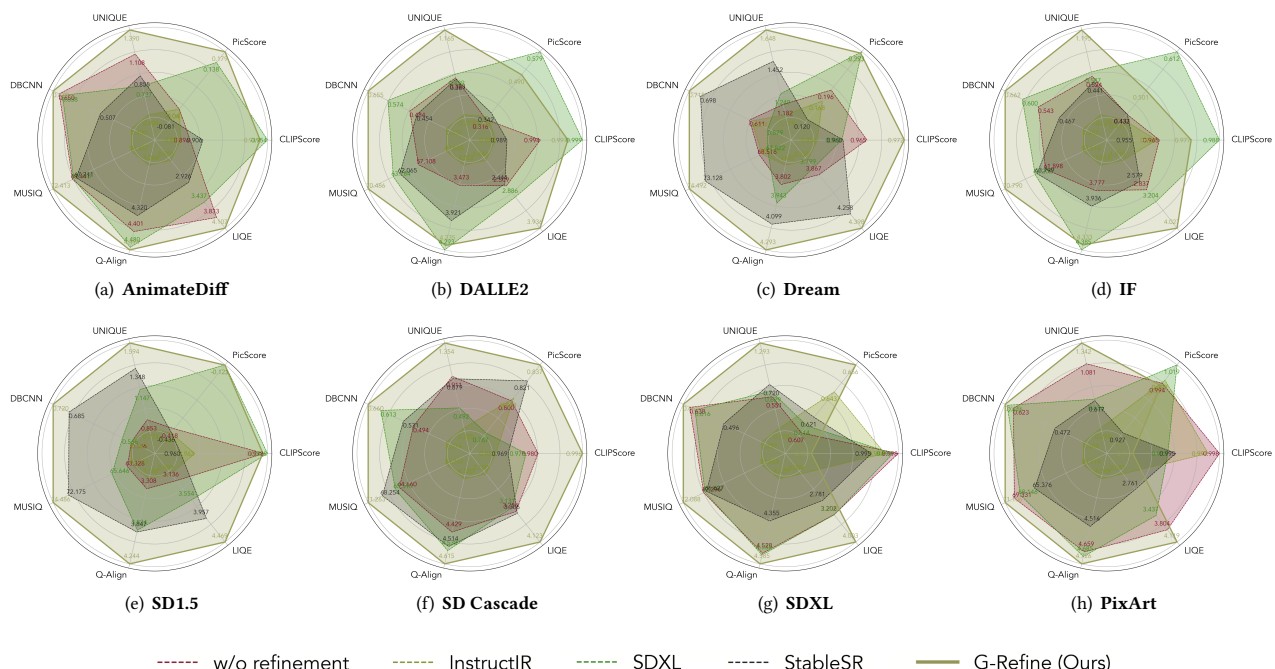

Figure 6: Radar maps for G-Refine on different original generative models.

**Table 6: Perceptual quality assessment result of PQ-Map and different indicators. [Key: Best; Below 0.6 ]**

| Database | AIGI-20K | | AGIQA-3K | | |
|---|---|---|---|---|---|
| Method | SROCC | PLCC | SROCC | PLCC | #Support Map? |
| DBCNN | **0.8506** | **0.8688** | 0.7488 | 0.7407 | ✗ |
| CLIPIQA | 0.7500 | 0.6375 | 0.6364 | 0.4518 | ✗ |
| CNN-IQA | 0.5968 | 0.5483 | 0.5913 | 0.5418 | ✗ |
| HyperIQA | 0.8223 | 0.5209 | 0.8407 | 0.4901 | ✗ |
| NIMA | 0.8466 | 0.7851 | **0.8764** | **0.7954** | ✗ |
| Paq2Piq | 0.1709 | 0.5030 | 0.2928 | 0.5727 | ✓ |
| **PQ-Map** (Proposed) | 0.7073 | 0.6910 | 0.7054 | 0.7084 | ✓ |

**Table 7: Alignment quality assessment result of AQ-Map and different indicators. [Key: Best; Below 0.6 ]**

| Database | AIGI-20K | | AGIQA-3K | | |
|---|---|---|---|---|---|
| Method | SROCC | PLCC | SROCC | PLCC | #Support Map? |
| CLIPScore | 0.4033 | 0.4903 | 0.4701 | 0.5341 | ✗ |
| ImageReward | 0.6113 | 0.6620 | 0.7298 | 0.7862 | ✗ |
| HPS | 0.5550 | 0.4971 | 0.6349 | 0.7000 | ✗ |
| HPSv2 | 0.6053 | 0.6385 | 0.6061 | 0.7164 | ✗ |
| PicScore | 0.5923 | 0.6106 | 0.6977 | 0.7633 | ✗ |
| CLIP Surgery | 0.4160 | 0.5225 | 0.5441 | 0.6648 | ✓ |
| **AQ-Map** (Proposed) | **0.6117** | **0.6797** | **0.7303** | **0.7862** | ✓ |

2 and excluded these components in Table 8. The result demonstrates the optimization effect, with Q-Align [37] and PicScore [13] representing perceptual/alignment quality respectively.

On SD1.5 with lower original quality, stage 1 plays a significant role in the optimization process. Conversely, on SD Cascade, which has a higher quality initially, the contribution of stage 1 is less pronounced and stage 2 becomes the primary driver of improvement.

When using only one quality map, they excel in enhancing perceptual or alignment quality individually, but their combined effect

**Table 8: Using G-Refine to optimize traditional and emerging generative models with different original quality. Abandoning PQ/AQ-Map as indicators, and deactivating Stage 2.**

| Database | | SD1.5 | | SD Cascade | |
|---|---|---|---|---|---|
| Indicator | Stage | PicScore | Q-Align | PicScore | Q-Align |
| PQ+AQ | 1,2 | **-0.1216** | **4.2436** | 0.8371 | **4.6149** |
| AQ | 1,2 | -0.1914 | 3.9463 | **0.8647** | 4.4621 |
| PQ | 1,2 | -0.2481 | 4.0368 | 0.8072 | 4.5983 |
| PQ+AQ | 1 | -0.2399 | 4.0409 | 0.8133 | 4.4690 |
| *Original Images* | | -0.4183 | 3.3082 | 0.8003 | 4.4294 |

on the other aspect is less effective. This highlights the importance of integrating both stages and utilizing both indicators for a comprehensive optimization of traditional and advanced T2I models, ensuring general optimization in perceptual/alignment quality.

## 5 CONCLUSION

In this study, we address the inconsistent generative quality of T2I models by proposing a quality-inspired general optimizer. Firstly, we enhance the CLIP's image and text encoders towards accurate perceptual quality maps for AIGIs. Secondly, we analyze prompts using a syntax tree, employing an ancestor tracing mechanism to yield alignment quality maps. Lastly, for precise and moderate optimization, these maps are employed to guide a multi-stage denoising process for AIGIS. These meticulously designed pipelines work in synergy to boost positive optimization for LQ while minimizing the negative impact on HQ images. Experimental results demonstrate that G-Refine improves AIGI's quality across 13 perceptual and alignment indicators and effectiveness to various T2I models, facilitating the adoption of T2I models in industrial production.

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
