# OpenReview forum: "G-Refine: A General Refiner for Text-to-Image Generation"
_acmmm.org/ACMMM/2024/Conference — MM2024 Oral_

### Official Review · Reviewer_2gJT · 2024-05-14

**Rating:** 4
**Confidence:** 4

**Summary:**

The paper "G-Refine: A General Quality Refiner for Text-to-Image Generation" introduces G-Refine, a model designed to enhance the quality of AI-generated images by addressing perceptual and alignment deficiencies. It integrates perceptual quality indicators (PQ-Map) and alignment quality indicators (AQ-Map) inspired by the Human Visual System, which identify and target low-quality regions without compromising high-quality areas. Through extensive experiments, the paper demonstrates that G-Refine significantly outperforms existing methods across multiple quality metrics, making AI-generated images more suitable for practical applications in various fields such as advertising, entertainment, and scientific research.

**Strengths:**

1) The insights on improving low-quality images in AIGIs tasks are very enlightening.

2) The novelty is good, with well-written text, beautiful figures, and a clear presentation of the problems and motivations.

3) The experimental results are good.

**Limitations:**

Need to reply:

1) Due to the complexity and abstraction of the three perceptual quality defects, it is challenging to ensure that the encoded information accurately represents these defects without additional supervisory information.

2) In equation (3), the interpolation method might introduce noise into the evaluation process of image perceptual quality. Although bi-cubic interpolation is relatively smooth, it can still produce artifacts or blurriness at edges or details.

3) How should the semantic ambiguity be considered in the impact on the Alignment Quality Indicator?

No mandatory to reply:

4) **I recommend the authors to open source their code**, as much of the theory and formulas in this paper seem "perfect", but in reality much of it is subjective and lacks interpretability, and while this is acceptable for a black-box approach to deep learning, it is essential that the code be made available to validate its credibility, given its good results. **I'll raise my score**, if the author can provide a link to the anonymized code to support their results.

5) Paper has somr typo errors, such as 'Figure 1 1' in lines 163-164 of the introduction section.

**Suitability:**

3

---

### Official Review · Reviewer_urTe · 2024-05-24

**Rating:** 4
**Confidence:** 3

**Summary:**

This paper proposes G-Refine to improve the quality of AI-generated images. It first identify a perceptual quality map and an alignment quality map, then apply a balanced-refiner to the LQ regions.

**Strengths:**

The paper is well-writen and easy to understand, the motivation is reasonable, and experimental results can support effectiveness of G-Refine.

**Limitations:**

1. In line 131, the author mentions that existing methods incur significant computational waste. However, no experimental results about the latency of the proposed method.
2. Decompose one single phrase "perceptual quality" to three phrases does not seem to make a big difference. Could you explain why you do so?

**Suitability:**

3

---

### Official Review · Reviewer_7A56 · 2024-05-25

**Rating:** 5
**Confidence:** 3

**Summary:**

The paper introduces G-Refine, a novel and comprehensive image quality enhancement framework designed to address the inconsistent generative quality in Text-to-Image (T2I) models. Extensive experiments demonstrate G-Refine's effectiveness, outperforming alternative optimization methods across multiple quality metrics and databases, thereby facilitating the broader adoption of T2I models in practical applications.

**Strengths:**

It presents a triad of interconnected modules: a perceptual quality indicator, an alignment quality indicator, and a general quality enhancement module.  These components synergize to identify and enhance low-quality regions in AI-generated images without compromising the integrity of high-quality areas.

**Limitations:**

There are my concerns:
1. The choice of perceptual prompts pairs is too subjective, there is a lack of ablation experiments.
2. Using tree-based methods for word matching may fail for more complicated and practical prompts. Sometimes, SD users write much longer prompts that describe the components at different levels. I wonder If when the prompts describe the overlap of areas will lead to the failure of the result.

**Suitability:**

3

---

### Official Review · Reviewer_brvx · 2024-05-26

**Rating:** 4
**Confidence:** 4

**Summary:**

This paper proposes an image enhancement framework called G-Refine.  The refine process is guided through two quality maps of perception and alignment.  The author's experimental data shows that G-Refine is suitable for a large number of T2I generative models and can promote the development and application of AIGC.

**Strengths:**

1. G-Refine contains multiple modalities.  For AIGC images, it optimizes T2I alignment for the first time. The joint treatment of two modalities is consistent with the theme of MM.
2. The quality maps in this article also has certain significance in the IQA task.  Beyond image enhancement tasks, they can be directly used as quality index.
3. The experiments include 4 verification data sets and 13 performance indicators, showing strong versatility.

**Limitations:**

1. The comparison algorithm used in this article is slightly inconsistent with the capitalization in the text and in the table.  Such as StableSR and InstructIR, etc.
2. The legend below Figure 6 is a bitmap, and the font is inconsistent with other parts.  It is recommended to unify.

**Suitability:**

2

---

### Meta-Review · Area_Chair_Nt2D · 2024-06-26

**Recommendation:** Accept (Oral)
**Confidence:** 5

**Metareview:**

Reviewers acknowledged the novelty and good-quality results of the paper. The rebuttal was successful, with two reviewers raising their scores, and the paper received all accept recommendations. The area chairs agree with this recommendation and are pleased to inform you that your paper has been accepted. The authors are advised to incorporate the reviewers' comments into the camera-ready version. Specifically, there is a remaining concern from Reviewer urTe wrt "perceptual quality" which needs clarification.